# Characterizing subgroups of sexual behaviors among men who have sex with men eligible for, but not using, PrEP in the Netherlands

**Feline de la Court**[1]*, **Daphne van Wees**[2], **Birgit van Benthem**[2], **Elske Hoornenborg**[1], **Maria Prins**[1,3], **Anders Boyd**[1,3,4]

**1** Department of Infectious Diseases, Public Health Service of Amsterdam, Amsterdam, The Netherlands, **2** Center for Infectious Disease Control, National Institute for Public Health and the Environment (RIVM), Bilthoven, The Netherlands, **3** Department of Infectious Diseases, Amsterdam Institute for Infection & Immunity (AII), Amsterdam UMC, University of Amsterdam, Amsterdam, The Netherlands, **4** Stichting HIV Monitoring, Amsterdam, The Netherlands

* fdlcourt@ggd.amsterdam.nl

**Data Availability Statement:** The data used in this study have been provided from the RIVM, where, due to ethical and legal restrictions, including the

## Abstract

This study identified subgroups of sexual behaviors associated with increased STI/HIV risk among those eligible for but not using pre-exposure prophylaxis (PrEP) in order to improve PrEP uptake and prioritization in the context of restricted capacity. We used data from sexual health centers (SHCs) in the Netherlands, including all visits of eligible but non-PrEP using men who have sex with men (MSM), men who have sex with men and women (MSMW) and transgender persons between July 2019 (start of the Dutch national PrEP pilot (NPP)) and June 2021. Using latent class analysis (LCA), we identified classes of sexual behaviors (number of partners, chemsex, group sex and sex work) and explored whether these classes were associated with STI diagnosis and sociodemographics. Across 45,582 visits of 14,588 eligible non-PrEP using individuals, the best fitting LCA model contained three classes of sexual behaviors. Classes were distinguished by seldomly reported sexual behaviors (class 1; 53.5%, n = 24,383), the highest proportions of ≥6 partners and group sex (class 2; 29.8%, n = 13,596), and the highest proportions of chemsex and sex work (class 3; 16.7% of visits, n = 7,603). Visits in classes 2 and 3 (vs. class 1) were significantly more often with individuals who were diagnosed with an STI, older (≥36 vs. ≤35 years), MSMW (vs. MSM), and visiting an urban (vs. non-urban) SHC; while these visits were significantly less often with individuals from an STI/HIV endemic area. The percentage of visits at which an STI was diagnosed was 17.07% (n = 4,163) in class 1, 19.53% (n = 2,655) in class 2 and 25.25% (n = 1,920) in class 3. The highest risk of STI, and thereby HIV, was in those engaging in specific subgroups of sexual behavior characterized by frequently reporting multiple partners, group sex, sex work or chemsex. PrEP uptake should be encouraged and prioritized for these individuals.

sensitivity of the information, data sharing is only allowed upon specific request and after approval by the registration committee. Please contact soap@rivm.nl for more information and/or submission requests.

**Funding:** The current study is part of the OptiPrEP (optimizing pre-exposure prophylaxis roll-out among men having sex with men) project, funded by the Aidsfonds (P-54601) to MP. The funders had no role in study design, data collection and analysis, decision to publish, or preparation of the manuscript.

**Competing interests:** MP obtained unrestricted research grants and speaker/ advisory fees from Gilead Sciences, Abbvie and MSD; all of which were paid to her institute and were unrelated to the current work. EH obtained unrestricted research grants from Gilead Sciences, which were paid to her institute and were unrelated to the current work. This does not alter our adherence to PLOS ONE policies on sharing data and materials. The other authors report no conflicts of interest.

# Introduction

Pre-exposure prophylaxis (PrEP) is a highly efficacious biomedical HIV prevention strategy [1]. In the Netherlands, PrEP implementation is targeted towards individuals with an increased risk of acquiring HIV, including men who have sex with men (MSM), men who have sex with men and women (MSMW) and transgender persons (TGP) [1, 2]. The current Dutch PrEP eligibility criteria pertain to MSM or TGP who have had (1) condomless insertive and/or receptive anal sex with a male partner with unknown HIV status, or with a known HIV-positive partner with a detectable viral load, (2) an anal sexually transmitted infection (STI), (3) syphilis, or (4) used post-exposure prophylaxis (PEP) in the past 6 months [2]. In 2019, a five-year national PrEP pilot (NPP) was implemented at sexual health centers (SHCs) in the Netherlands, offering subsidized PrEP to 8,500 eligible users [3]. However, because the provision capacity of the NPP is restricted [3], it remains unclear whether there are deficiencies in PrEP uptake among certain subgroups of eligible individuals. Identifying and characterizing eligible non-PrEP users with the highest PrEP need is thus crucial.

In addition to the Dutch PrEP eligibility criteria, certain sexual behaviors are associated with an increased risk of engaging in condomless anal sex and acquiring an HIV infection, which include having multiple partners [4], chemsex [5], group sex [6], and sex work [7, 8]. Previous studies have shown that chemsex [9] and sex work [10] are also associated with lower PrEP uptake; as are certain sociodemographic characteristics, such as younger age (i.e., <30 or <35), living in a non-urban area, lower education level (i.e., no post-secondary education), or having a migration background from an HIV/STI endemic area [11–13]. Such behaviors and characteristics can help more effectively recognize PrEP need and prioritize individuals for the NPP beyond the current eligibility criteria. This study therefore aimed to identify subgroups of sexual behavior among MSM, MSMW and TGP who were eligible for, but did not use, PrEP between July 2019 and June 2021. Furthermore, we aimed to explore whether these subgroups were associated with STI diagnosis and sociodemographic variables. Considering that many of the sexual behaviors are highly correlated, making it difficult to construct models examining independent pathways to these outcomes, we used an approach that clustered like behaviors within individuals as latent classes to accomplish these study aims.

# Methods

## Study design

Data from the Dutch national STI and HIV surveillance database were used. Data are collected from a nationwide system of 24 public sexual health centers (SHCs) across 9 regions in the Netherlands [14]. Free-of-charge STI/HIV testing and care is offered at these centers, which are targeted towards populations at higher risk for STI or HIV infection [14]. Collected data pertain to STI and HIV testing and diagnosis, self-reported sexual behavior, and additional sociodemographic and sexual health-related characteristics. Use of data was requested for analysis in July 2021, which pertains to data collected from July 2019 until June 2021.

All individuals visiting an SHC were of age and provided both verbal informed consent and an opt-out option for sharing data with the RIVM, documented by the SHC. All collected data are coded, secured and fully pseudonymized in accordance with Dutch privacy legislation.

## Study population

Between July 2019 and June 2021, we included all visits of HIV-negative MSM, MSMW and TGP with at least one visit at an SHC and who did not report any PrEP use in the preceding 12

months. We further included only visits from individuals who met at least one PrEP eligibility criterion according to the Dutch guidelines [2].

## Study variables

Sexual behavior characteristics were the number of partners, chemsex (defined as cocaine, ketamine, mephedrone, gamma-hydroxybutyrate, gamma-butyrolactone, and/or crystal meth use around the time of or during sex [15]), group sex and sex work. Additional variables were STI (defined as anal chlamydia, anal gonorrhea, hepatitis C virus, hepatitis B virus, and/or syphilis diagnosis) and sociodemographic variables, including age (younger/older; cut-off based on median age), education level (low-middle/high; low-middle = primary, secondary, vocational or specialist education, and high = associate, bachelor, master or doctoral degree [16]), sexual partner(s) (male/male and female), originating from an STI/HIV endemic area (yes/no; defined according to the National Institute for Public Health and the Environment (RIVM) as being born in or having either one or both parents born in Suriname, Turkey, Netherlands Antilles, North Africa, Sub-Saharan Africa, Eastern Europe, Central and South America, or Asia [3, 17]) and region of SHC visit (urban/non-urban; where urban is all "Randstad" provinces and non-urban is all other provinces). All variables refer to the six months prior to each visit except for any STI, which pertains to STI diagnosed at the current visit.

## Statistical analysis

Descriptive statistics were calculated for sexual behavior and both STI and sociodemographic variables. Latent class analysis (LCA) was performed to distinguish classes of sexual behavior that could be targeted for PrEP uptake. The sexual behaviors examined in LCA were number of partners, chemsex, group sex and sex work in the past six months.

We used a generalized structural equation modelling approach [18]. Briefly, a latent variable model was constructed, based solely on sexual behavior variables, where the probability of engaging in each behavior given latent class $k$ was modelled by an intercept, $\alpha_{dk}$, specific to each sexual behavior variable $d$ and class $k$. Sexual behavior characteristics were modeled as dichotomous variables: number of partners ($\geq 6/\leq 5$; cut-off based on median), chemsex (yes/no), group sex (yes/no) and sex work (yes/no). Models were estimated using maximum likelihood, calculated by summing all conditional likelihoods of each latent class multiplied by the associated latent class probabilities. The posteriori probability of a visit $i$ belonging to each class $k$, $\pi_{ik}$, was determined from this likelihood. Visits were then assigned a latent class $k$ corresponding to the highest probability $\pi_{ik}$. We examined participant characteristics of each assigned class across visits. The conditional probabilities of each item (i.e., sexual behaviors) were estimated with intercept-only logit models for each item within classes.

We determined the number of latent classes (1 to 6) using the Bayesian Information Criterion value (BIC) and Akaike Information Criterion (AIC) score, for which lower values indicate a better fit, and an entropy calculation ranging from 0–1, where a higher value indicates higher ability of the model to classify clusters (i.e., degree of class membership separation) [19]. Using the best fitting model, the univariable odds ratios (ORs) and 95% confidence intervals (CIs; using the delta method) for the associations between STI diagnosis or sociodemographic variables (i.e., STI, number of partners, chemsex, group sex, sex work, age, education level, sexual partner(s), originating from an STI/HIV endemic area, and region of SHC visit) and latent classes were obtained from the modeled parameter estimates. Multivariable ORs were then calculated from the parameter estimates of a model including all covariates. Variance estimates were corrected for repeated observations within individuals using a clustered sandwich estimator.

Sensitivity analyses were performed to assess potential bias related to calendar year and to the inclusion of multiple visits per individual. The LCA model was hence repeated (i) for each year, separately, and (ii) using only one randomly selected visit per individual.

For all analyses, STATA (v16, College Station, TX, USA) statistical software was used. Latent class models were estimated using the "gsem" command, and "predict" post-estimation commands were used for posteriori probabilities.

## Results

### Study population

From the 103,216 visits that took place at SHCs from July 2019 to June 2021 among non-PrEP using MSM who were HIV-negative at first visit in the study period, we excluded 253 visits for the 250 individuals who became HIV positive from the moment of diagnosis, and 57,381 visits at which individuals were not eligible for PrEP. In total, 14,588 individuals were included in analyses, contributing 45,582 visits (median visits per individual = 2; IQR = 1–4).

At first visit between July 2019 to June 2021 (Table 1), the majority was MSM (98.5%), a small proportion was TGP (1.5%), and of the MSM, around 17.7% were MSMW. At this visit, most individuals were aged ≤35 years (59.2%), had a high level of education (59.0%), were not from an STI/HIV endemic area (83.1%), and were visiting an SHC in an urban area (65.7%).

### Classes of sexual behavior

LCA revealed three different classes of sexual behaviors at which individuals were eligible for but did not use PrEP between July 2019 and June 2021. This 3-class LCA model showed the best fit based on higher entropy, and lower BIC and AIC compared to models with more or less classes (S1 Table). Class 1 was characterized by a low number of partners (≤5) and infrequent reports of recent chemsex, group sex and sex work, all in the 6 months prior to the visit (Fig 1). Class 2 was characterized by a large proportion of visits at which a high number of partners (≥6) and group sex were reported. Class 3 was characterized by a high number of visits at which numerous partners (≥6), chemsex and group sex were reported, along with relatively more frequent reports of sex work compared to class 1 and 2.

The proportion of visits assigned to class 1, 2 and 3, based on the highest class membership probability, were 53.5% (n = 24,383), 29.8% (n = 13,596) and 16.7% (n = 7,603), respectively (Table 2). The degree of class separation (i.e., entropy) was 0.79, suggesting fairly high ability of the model to classify clusters [19]. The distributions of posteriori probabilities according to the assigned class memberships are provided in S1 Fig.

Sensitivity analyses stratifying by year to indicate potential longitudinal trends show very similar proportions of sexual behaviors over time with some small deviations in classes 2 and 3 (S2 Fig and S3 Table). Moreover, the class membership sizes (i.e., the number of visits assigned to each class) were quite stable in 2019 and 2020, but in 2021, the number of visits were notably higher in class 1 and lower in class 2.

### STI diagnosis and sociodemographic variables associated with sexual behavior classes

An STI was diagnosed at 19.2% (n = 8,738) of all included visits. Per class, the percentage of visits at which an STI was diagnosed was 17.07% (n = 4,163) in class 1, 19.53% (n = 2,655) in class 2 and 25.25% (n = 1,920) in class 3. Table 3 shows that in multivariable analysis, with class 1 as the reference group, the visits classified in class 2 and class 3 were significantly more often with individuals who were diagnosed with an STI, older (≥36 vs. ≤35 years), MSMW

**Table 1. Study population characteristics stratified by class membership at first visit between July 2019 and June 2021.**

| | | Classes | | |
|---|---|---|---|---|
| | **Total** | **Class 1** | **Class 2** | **Class 3** |
| | **n = 28,739** | **n = 16,455** | **n = 8,195** | **n = 4,089** |
| | **% (n)** | **% (n)** | **% (n)** | **% (n)** |
| Age, years | | | | |
| ≤35 years | 59.2 (17,007) | 63.34 (10,422) | 53.73 (4,403) | 53.36 (2,182) |
| ≥36 years | 40.8 (11,732) | 36.66 (6,033) | 46.27 (3,792) | 46.64 (1,907) |
| Sex/Gender | | | | |
| Male | 98.5 (28,310) | 98.92 (16,277) | 97.99 (8,030) | 97.90 (4,003) |
| Transgender | 1.5 (429) | 1.08 (178) | 2.01 (165) | 2.10 (86) |
| Sexual partner(s) ˙ | | | | |
| Male | 82.3 (23,637) | 83.62 (13,759) | 81.78 (6,702) | 77.67 (3,176) |
| Male and female | 17.7 (5,084) | 16.33 (2,687) | 18.15 (1,487) | 22.25 (910) |
| Region* | | | | |
| Non-urban | 34.3 (9,847) | 37.16 (6,114) | 29.81 (2,443) | 31.55 (1,290) |
| Urban | 65.7 (18,892) | 62.84 (10,341) | 70.19 (5,752) | 68.45 (2,799) |
| From an STI/HIV endemic area ** ˙˙ | | | | |
| No | 78.9 (22,669) | 78.63 (12,938) | 78.19 (6,408) | 81.27 (3,323) |
| Yes | 20.9 (6,014) | 21.20 (3,488) | 21.53 (1,764) | 18.64 (762) |
| Education level ˙˙˙ | | | | |
| Low-middle | 32.81 (9,429) | 33.56 (5,522) | 29.38 (2,408) | 36.66 (1,499) |
| High | 59.01 (16,960) | 58.87 (9,687) | 61.28 (5,022) | 55.05 (2,251) |

Explanation of data: Models were estimated using maximum likelihood, which was calculated by summing all conditional likelihoods of each latent class multiplied by the associated latent class probabilities. The posteriori probability of a visit $i$ belonging to each class $k$, $\pi_{ik}$, was determined from this likelihood. Visits were then assigned a latent class k corresponding to the highest probability $\pi_{ik}$. Data are presented as percentages (n). Abbreviations: CAS = condomless anal sex; PEP = post exposure prophylaxis. Education level is defined as: low-middle = primary, secondary, vocational or specialist education; and high = associate, bachelor, master or doctoral degree.

*Region is defined as urban, referring to all "Randstad" provinces, or non-urban, referring to all other provinces.

**Originating from an STI/HIV endemic area is defined as being born in and having either one or both parents born in Surinam, Turkey, Netherlands Antilles, North Africa, Sub-Saharan Africa, Eastern Europe, Central and South America, or Asia.

˙ Missing values for "sexual partner(s)": total = 18 (0.06%); class 1 = 9 (0.05%); class 2 = 6 (0.07%); class 3 = 3 (0.07%).

˙˙ Missing values for "from an STI/HIV endemic area": total = 56 (0.19%); class 1 = 29 (0.18%); class 2 = 23 (0.28%); class 3 = 4 (0.10%).

˙˙˙ Missing values for "education level": total = 1,246 (7.57%); class 1 = 1,353 (7.37%); class 2 = 765 (9.33%); class 3 = 339 (8.29%).

(vs. MSM), and visiting an SHC in an urban (vs. non-urban) area. The odds ratios for being MSMW and visiting an SHC in an urban area were both notably high. Furthermore, the visits classified in classes 2 and 3, compared to those classified in class 1, were significantly less often with individuals who were from an STI/HIV endemic area. The visits in class 2 compared to class 1 had a significantly higher odds of being with individuals with a high (vs. low-middle) level of education. When comparing class 2 and 3 (S2 Table), the visits in class 3 (vs. class 2) were significantly more often with individuals diagnosed with an STI, aged ≥36 years and from an STI/HIV endemic area; while the odds of these visits were significantly lower in individuals who were MSMW (vs. MSM) and visiting an SHC in an urban (vs. non-urban) area.

In a sensitivity analysis, where the LCA model was repeated using only one randomly selected visit per individual, results were comparable to the original model (S4 Table); indicating that inclusion of multiple visits per participant did not likely bias results.

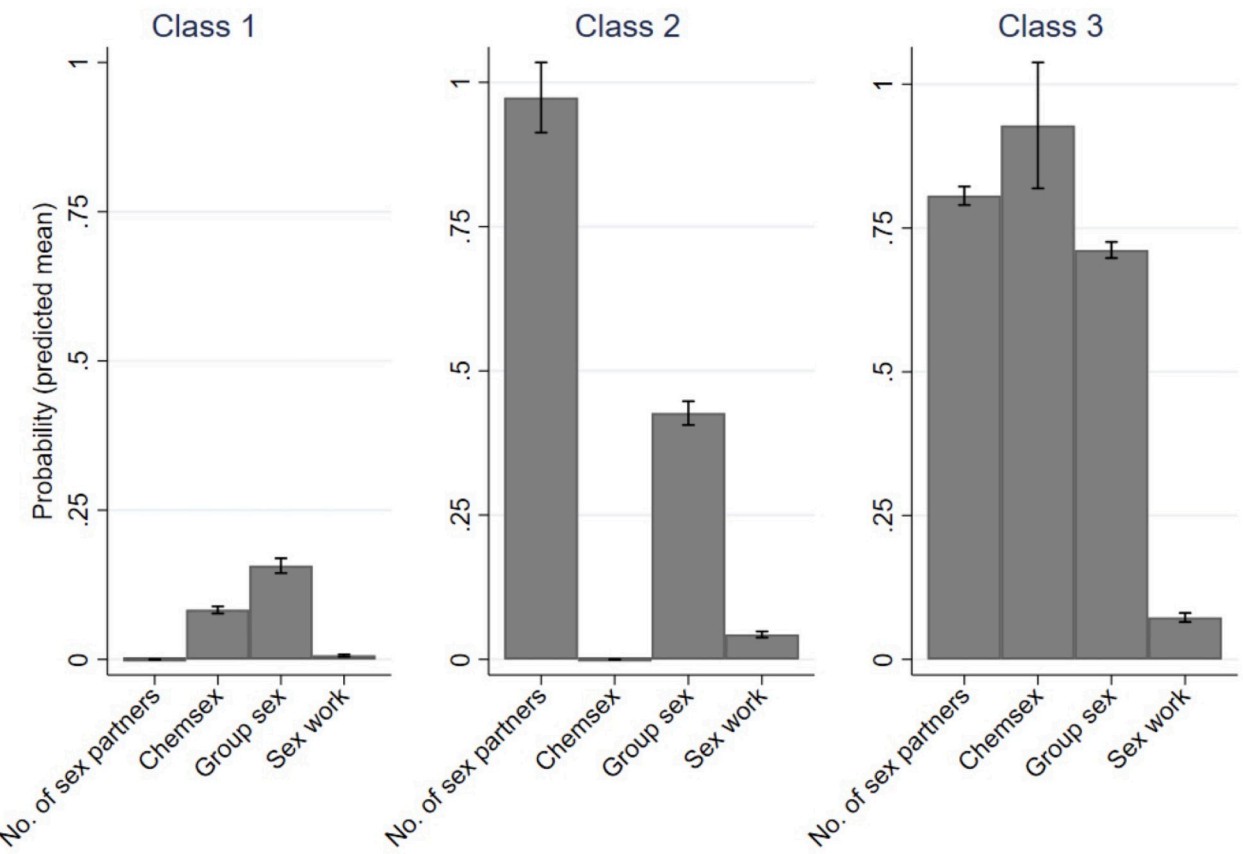

**Fig 1. Sexual behavior associated with increased STI risk across three latent classes.** Explanation of data: Bars represent the mean proportion of visits reporting each sexual behavior respectively for class 1, 2 and 3. All sexual behaviors refer to the six months prior to the visit. Number (= No.) of sexual partners refers to those with ≥6 sexual partners in the six months prior to the visit. Chemsex was defined as using cocaine, ketamine, mephedrone, gamma-hydroxybutyrate (GHB), gamma-butyrolactone (GBL), and/or crystal meth around or during sex. Bands at the top of each bar represent 95% confidence intervals, which were calculated using the delta method.

**Table 2. Predicted probabilities (in percentages) of class membership and the outcome variables within each latent class.**

|  | Class 1 | Class 2 | Class 3 |
|---|---|---|---|
|  | n visits = 24,383 | n visits = 13,596 | n visits = 7,603 |
|  | % | % | % |
| **Class size*** | 53.5 | 29.8 | 16.7 |
|  | % (95% CI) | % (95% CI) | % (95% CI) |
| **Number of partners ≥6** | 0 | 97.4 (77.6–99.8) | 80.6 (79.0–82.2) |
| **Chemsex**** | 8.3 (7.7–8.9) | 0 | 92.8 (71.4–98.5) |
| **Group sex** | 15.7 (14.5–17.0) | 42.7 (40.6–44.7) | 71.2 (69.8–72.6) |
| **Sex work** | 0.7 (0.5–0.9) | 4.3 (3.8–4.9) | 7.3 (6.5–8.1) |

Explanation of data: Results are presented as marginal predicted means of the outcome within each latent class, as obtained by the "*lcmean*" command in STATA. Data are presented as % = percentage; 95% CI = 95% confidence interval.

*Class size indicates the size of each class in percentages after visits were assigned to their most likely class membership.

**Chemsex was defined as using cocaine, ketamine, mephedrone, gamma-hydroxybutyrate (GHB), gamma-butyrolactone (GBL), and/or crystal meth around or during sex.

**Table 3. Association between class membership and various factors (LCA model with covariates) comparing classes 2 and 3 to class 1.**

| | Class 2 | | | | Class 3 | | | |
| | Vs. class 1 | | | | Vs. class 1 | | | |
| | OR | 95%CI | aOR* | 95%CI | OR | 95% CI | aOR* | 95%CI |
|---|---|---|---|---|---|---|---|---|
| **Any STI**** Yes vs. no | 1.1 | 1.0–1.2 | 1.3 | 1.1–1.5 | 1.9 | 1.7–2.1 | 1.8 | 1.6–2.0 |
| **Age** ≥36 vs. ≤35 years | 1.3 | 1.2–1.5 | 1.3 | 1.2–1.5 | 2.2 | 2.0–2.4 | 2.4 | 2.2–2.6 |
| **Sexual partner(s)** Male and female vs. male | 1.1 | 1.0–1.2 | 2.2 | 1.8–2.6 | 1.4 | 1.3–1.5 | 1.4 | 1.2–1.8 |
| **Region**** Urban vs. non-urban | 1.5 | 1.4–1.6 | 1.9 | 1.7–2.2 | 1.4 | 1.3–1.6 | 1.9 | 1.7–2.2 |
| **From an STI/HIV endemic area***** Yes vs. no | 1.0 | 0.9–1.1 | 0.7 | 0.7–0.8 | 0.8 | 0.7–0.9 | 0.8 | 0.7–0.9 |
| **Education level** High vs. low-middle | 1.3 | 1.2–1.4 | 1.2 | 1.1–1.3 | 0.9 | 0.8–1.0 | 1.0 | 0.9–1.1 |

Parameter estimates for the associations between STI diagnosis or sociodemographic variables and latent classes were directly obtained from a generalized structural equation model. Explanation of data: OR = odds ratio; aOR = adjusted odds ratio; 95% CI = 95% confidence interval.

*aOR: all models were adjusted for the variables present in the table.

**Any STI includes anal chlamydia, anal gonorrhea, hepatitis C virus, hepatitis B virus, and syphilis diagnosed at the visit.

***Originating from an STI/HIV endemic area is defined as being born in and having either one or both parents born in Surinam, Turkey, Netherlands Antilles, North Africa, Sub-Saharan Africa, Eastern Europe, Central and South America, or Asia.

****Region is defined as urban, referring to all "Randstad" provinces, or non-urban, referring to all other provinces.

## Discussion

This study has identified three latent classes of sexual behaviors among MSM, MSMW and TGP who are eligible for, but not using, PrEP. Most visits belonged to a subgroup of sexual behavior with a lower number of partners, and few reports of group sex, chemsex and sex work (i.e., class 1). Other visits belonged to subgroups of sexual behavior involving a higher number of partners and reported group sex (i.e., class 2), and to a lesser extent, of chemsex and sex work while also having numerous reports of high numbers of partners, group sex and sex work (i.e., class 3). The proportion of STI diagnosis increases with higher class number, suggesting that these subgroups represent one mean of assessing risk for acquiring STI, and thus also for HIV.

Individuals of all three classes were technically eligible for PrEP. However, the largest group, class 1, had a much lower proportion with STIs, likely related to different sexual behaviors compared to classes 2 and 3. Given that STI has been established as an important proxy for HIV infection [20, 21], individuals who exhibit the behaviors in class 1 at a given visit could be less prioritized for PrEP, despite being eligible, in favor of individuals who exhibit the behaviors in class 2 and 3. The identified risk groups based on sexual behavior profiles may therefore be more informative than simply PrEP eligibility in allocating PrEP to those at highest risk for HIV. In a situation of restricted subsidized PrEP provision, such as in the Netherlands, individuals presenting to the SHCs with sexual behaviors in line with class 3 should certainly be prioritized for PrEP.

When focusing on the components of classes 2 and 3, having more partners and engagement in group sex are predominant features of these latent classes and are both associated with increased HIV risk [4, 6]. One noteworthy difference between classes 2 and 3 is that at almost

all visits in class 3, chemsex was reported. Even though chemsex is not in itself a transmission factor for HIV, it represents an environmental factor in which individuals are more at risk of exposure to HIV and condomless anal sex [22]. This finding is in line with a previous study using LCA in which classes including chemsex were associated with increased HIV and STI positivity among all MSM attending sexual health centers (SHC) in the Netherlands [23]. It should also be noted that specific combinations of chemsex drugs, such as erectile dysfunction drugs with nitrites and polydrug use, might be more closely linked to STI risk [24]. Moreover, we have limited data on whether chemsex includes injecting drug use due to substantial missing data. However, because it poses an increased risk for HIV acquisition, it is worth mentioning as another behavioral characteristic to consider for PrEP uptake [25]. Lastly, sex work was also an important feature for class 3; however, it was uncommon.

Individuals who presented at an SHC and belonged to class 2 or 3 were significantly more often 36 years old or older, MSM, not from an STI/HIV endemic area and visiting urban SHCs when compared to class 1. Notably, the demographic characteristics differentiating class 3 versus 2 were higher age (≥36 years old), lower proportion of MSMW, less likely to visit an urban SHC and more likely to be from an STI/HIV endemic area. As our data from the SHCs includes a noticeable proportion of individuals from an STI/HIV endemic area and non-urban SHC visitors, it can be assumed that although these individuals are not using PrEP, they are seeking care for their sexual health. Given their increased risk of STIs, as shown especially by the association with class 3, PrEP provision needs to be discussed and offered to these individuals from STI/HIV endemic countries and in non-urban areas [12, 26]; either within or outside of the NPP. Individuals from STI/HIV endemic regions were also more present in group 1 compared to other groups. This is despite their potential risk for HIV, as shown in research indicating that the majority of HIV acquisition among MSM migrants from Sub-Saharan Africa and Latin America/Caribbean are postmigration HIV infections, acquired in the European host country [27]. Nonetheless, those from STI/HIV endemic regions that were more often classified in group 1, represents individuals from heterogenous geographical locations. Given that we did not have access to specific information on the origin of these individuals or the lower risk of STI/HIV in the group with a larger proportion of migrants from endemic regions, this becomes difficult to interpret but important to consider for future research.

It is unknown whether the lack of PrEP use found among the eligible MSM reporting high proportions of sexual behaviors associated with STI/HIV (i.e., class 2 and 3) in our study may be attributed to not wanting PrEP, institutional rather than behavioral factors, such as being waitlisted for the NPP, or both. Interestingly, our analyses indicate that over the years, the proportion of visits increased in class 1, deceased in class 2, and remained stable in class 3. This shift in class membership is difficult to explain but could be due to changes in sexual behaviors due to COVID-19 restrictions [28, 29], the enrolment of individuals with high risk of HIV into the NPP over time, or other factors. The potential barriers for PrEP uptake may be alleviated by improving PrEP awareness, knowledge and providing alternative routes of affordable PrEP provision outside of the NPP such as telehealth. Further (qualitative) research is needed to gain a deeper understanding of why those in classes 2 and 3 are not using PrEP despite eligibility, patterns of PrEP uptake in relation to behavior and to what extent personal and institutional barriers play a role.

Our findings are based on a large sample size, using extensive surveillance data at a national level. Also, our data pertain to only visits at SHC. Even though this excludes those seeking sexual healthcare elsewhere, SHC visitors importantly reflect a population that is generally at increased risk of STI and HIV [3], thus for whom PrEP would more likely be indicated. Moreover, relatively few studies among general MSM populations report on those who do sex work, despite the importance of HIV prevention for this group. Lastly, because we have focused our

study on behavior beyond the scope of the current PrEP eligibility criteria, we provide a more targeted way of allocating PrEP to those who need it most based on specific sexual behaviors and sociodemographic characteristics; especially when subsidized provision capacity is restricted.

Our study is limited in that we could not compare non-PrEP users to PrEP users. Previous research shows that these two groups did not differ sociodemographically, but PrEP users had more STIs, more sex partners, CAS and chemsex than non-PrEP users [30]. Because this comparison was not possible in our study, it is difficult to determine how the identified non-PrEP users differ from PrEP users and whether classes observed in this study are different among PrEP users. However, we do see that the STI prevalence is higher in classes 2 (20%) and 3 (25 = %) than among PrEP users in the NPP (17%) [3], indicating that those in classes 2 and 3 may have a similar or even higher need for PrEP than the average PrEP user in the NPP when considering the association between STI and HIV risk [20, 21].

## Conclusions

In conclusion, this study has identified classes of sexual behavior that are associated with increased risk of STI and, as a strong correlate thereof, HIV. It is concerning that the individuals belonging to these classes who are visiting an SHC, are not using PrEP despite being eligible for it. We found that when many sexual partners, group sex, sex work or chemsex are reported, extra attention to encourage PrEP uptake is warranted. Prioritizing individuals with one or more of these characteristics may be attained through targeted information provision and counseling. In the case of restricted access and waitlists, an assessment for PrEP need based on the combination of these characteristics may be beneficial.

## Supporting information

**S1 Table. Comparison of fit statistics for latent class analysis with 1–6 classes.** Abbreviations: AIC = Akaike's information criterion; BIC = Bayesian information criterion *Entropy could not be calculated.
(DOCX)

**S2 Table. Association between class membership and various factors (LCA model with covariates) comparing class 2 and 3.** Parameter estimates for the associations between STI diagnosis or sociodemographic variables and latent classes were directly obtained from a generalized structural equation model. Explanation of data: OR = odds ratio; aOR = adjusted odds ratio; 95% CI = 95% confidence interval. *aOR: all models were adjusted for the variables present in the table. **Any STI includes anal chlamydia, anal gonorrhea, hepatitis C virus, hepatitis B virus, and syphilis diagnosed at the visit. ***Originating from an STI/HIV endemic area is defined as being born in and having either one or both parents born in Suriname, Turkey, Netherlands Antilles, North Africa, Sub-Saharan Africa, Eastern Europe, Central and South America, or Asia. ****Region is defined as urban, referring to all "Randstad" provinces, or non-urban, referring to all other provinces.
(DOCX)

**S3 Table. Class membership size (number of visits) across three latent classes, stratified by year (2019, 2020 and 2021).** Explanation of data: Models were estimated using maximum likelihood, which was calculated by summing all conditional likelihoods of each latent class multiplied by the associated latent class probabilities. The posteriori probability of a visit $i$ belonging to each class $k$, $\pi_{ik}$, was determined from this likelihood. Visits were then assigned a latent class

*k* corresponding to the highest probability $\pi_{ik}$. Data are presented as percentages (n).
(DOCX)

**S4 Table. Association between class membership and various factors (LCA model with covariates) comparing class 2 and 3 to class 1; using one randomly selected visit per individual.** Parameter estimates for the associations between STI diagnosis or sociodemographic variables and latent classes were directly obtained from a generalized structural equation model. Explanation of data: OR = odds ratio; aOR = adjusted odds ratio; 95% CI = 95% confidence interval. *aOR: all models were adjusted for the variables present in the table. **Any STI includes anal chlamydia, anal gonorrhea, hepatitis C virus, hepatitis B virus, and syphilis diagnosed at the visit. ***Originating from an STI/HIV endemic area is defined as being born in and having either one or both parents born in Surinam, Turkey, Netherlands Antilles, North Africa, Sub-Saharan Africa, Eastern Europe, Central and South America, or Asia. ****Region is defined as urban, referring to all "Randstad" provinces, or non-urban, referring to all other provinces.
(DOCX)

**S1 Fig. The a posteriori probabilities of class membership per class.** Explanation of data: models were estimated using maximum likelihood, which was calculated by summing all conditional likelihoods of each latent class multiplied by the associated latent class probabilities. The posteriori probability of a visit *i* belonging to each class *k*, $\pi_{ik}$, was determined from this likelihood. Visits were then assigned a latent class *k* corresponding to the highest probability $\pi_{ik}$. The figures show the distribution of probabilities for belonging to a class given the assigned class membership. Each point represents an individual consultation visit. Figure A shows the probabilities for those assigned to class 1. Figure B shows the probabilities for those assigned to class 2. Figure C shows the probabilities for those assigned to class 3. For example, the dots in the figure indicate that only few visits had a lower probability of belonging to a given class, and the longer lines (subsequent dots), indicate that the majority of visits had (almost) 100% probability of belonging to a given class, and a very low probability of belonging to another latent class.
(DOCX)

**S2 Fig. Sexual behavior associated with increased STI risk across three latent classes, stratified by year (2019, 2020 and 2021).** Explanation of data: Bars represent the mean proportion of visits reporting each sexual behavior respectively for class 1, 2 and 3. All sexual behaviors refer to the six months prior to the visit. Number (= No.) of sexual partners refers to those with ≥6 sexual partners in the six months prior to the visit. Chemsex was defined as using cocaine, ketamine, mephedrone, gamma-hydroxybutyrate (GHB), gamma-butyrolactone (GBL), and/or crystal meth around or during sex. Bands at the top of each bar represent 95% confidence intervals, which were calculated using the delta method.
(DOCX)

## Acknowledgments

The authors gratefully acknowledge the nurses, physicians and supporting staff at the centres for sexual health, and the data managers and researchers at the RIVM for their contribution to the data collection.

## Author Contributions

**Conceptualization:** Feline de la Court, Daphne van Wees, Maria Prins, Anders Boyd.

**Formal analysis:** Feline de la Court.

**Funding acquisition:** Maria Prins.

**Methodology:** Feline de la Court, Daphne van Wees, Anders Boyd.

**Project administration:** Feline de la Court, Maria Prins.

**Supervision:** Daphne van Wees, Maria Prins, Anders Boyd.

**Visualization:** Feline de la Court.

**Writing – original draft:** Feline de la Court.

**Writing – review & editing:** Feline de la Court, Daphne van Wees, Birgit van Benthem, Elske Hoornenborg, Maria Prins, Anders Boyd.

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
