## [Decision Letter · Decision Letter 0]

2 Jan 2023

PONE-D-22-30781Characterizing subgroups of sexual behaviors among men who have sex with men eligible for, but not using, PrEP in the NetherlandsPLOS ONE

Dear Dr. de la Court,

Thank you for submitting your manuscript to PLOS ONE. After careful consideration, we feel that it has merit but does not fully meet PLOS ONE’s publication criteria as it currently stands. Therefore, we invite you to submit a revised version of the manuscript that addresses the points raised during the review process.

 Both reviewers agree that your manuscript needs some changes before it can be accepted. Please address all their suggestions before resubmitting.  I particularly share the concern by one of the reviewers that your analysis could have been biased the fact that multiple observations (visits) were included for the same participant. I suggest to conduct some kind of sensitivity analysis, e.g. by only using one random visit from each study participant. Also as suggested by the reviewer, please also provide results from a "traditional" risk factor approach (e.g. using logistic regression with any STI as the outcome while adjusting for multiple observations per participant) for complementing the LCA approach. ==============================

We look forward to receiving your revised manuscript.

Kind regards,

Joël Mossong, PhD

Academic Editor

PLOS ONE

Journal Requirements:

"MP obtained unrestricted research grants and speaker/ advisory fees from Gilead Sciences, Abbvie and MSD; all of which were paid to her institute and were unrelated to the current work. EH obtained unrestricted research grants from Gilead Sciences, which were paid to her institute and were unrelated to the current work. The other authors report no conflicts of interest."

Reviewers' comments:

Reviewer's Responses to Questions

**Comments to the Author**

1. Is the manuscript technically sound, and do the data support the conclusions?

Reviewer #1: Yes

Reviewer #2: Yes

2. Has the statistical analysis been performed appropriately and rigorously? 

Reviewer #1: Yes

Reviewer #2: Yes

3. Have the authors made all data underlying the findings in their manuscript fully available?

Reviewer #1: Yes

Reviewer #2: No

4. Is the manuscript presented in an intelligible fashion and written in standard English?

Reviewer #1: Yes

Reviewer #2: Yes

5. Review Comments to the Author

Reviewer #1: Very well written paper, very concise.

I only have very few comments.

Introduction:

- last paragraph before method section: You aim to identify subgroups of MSM, MSMW and TGP eligible for but not using PrEP. What I was missing here was migrants from endemic areas. I suppose that also in the NL there are many migrants from SubSaharan Africa that do not fall in any of the categories above and still carry a high risk for HIV (i.e. by having heterosexual sex with migrants from same region). Why were they not taken into account? Do they not attend the SHC?

Are iv drug users not considered a group needing PrEP? Are they in a different health setting?

Methods:

- page 4, 1st paragraph: was the sexual behaviour self-reported or reported by physicians?

- Page 5, last paragraph: why did you exclude the period of summer 2021?

Results/Discussion:

- Again, people from STI/HIV endemic regions are more present in group 1 and thus deprioritised. Although this is what the model is sugesting I would have liked to see this critically discussed in more depth. People from STI/HIV endemic areas are a very heterogenous group and I doubt that they all carry the same high risk for HIV. You do not necessarilöy have many STI in this group and still, they may have a high risk for HIV with partners from endemic regions.

Among these, are there any migrant subgroups that are bigger than others, have a higher risk for HIV? If so, I wonder if you have tried to look at them separately?

- Page 12, last paragraph:

"We found that when many sexual partners, group sex, sex work and chemsex are reported, extra attention to encourage PrEP uptake is warranted" - Should it not be "sex work OR chemsex"? I understood that one of the cited factors would be sufficient to warrant stronger encouragement to uptake PrEP. I suppose it is quite rare to have all factors present in one person.

- It would have been nice to make more concrete recommendations on how to prioritise the groups. How should counsellors proceed to make a concise choice opn who to approach? Are you recommending a scoring system?

- Please mention what would be foreseen what iv drug users.

Reviewer #2: INTRODUCTION

1. Given the abundant existing literature on PrEP uptake (and non-uptake), the introduction could benefit from a clearer justification for using an LCA approach, as opposed to say, a traditional risk factor analysis. What new or additional insights do the authors expect to gain using this technique?

METHODS

2. In this reviewer’s understanding, LCA models decipher latent structure underlying a population, not their visit patterns. Therefore the authors’ decision to include data from multiple visits from the same subjects seems as though it would falsely inflate the dataset. This likely over represents the experiences of those with more study visits, weighting results that skew towards their characteristics.

3. A clearer justification for the authors’ decision to code for whether a participant is from an HIV endemic setting would be helpful. This could include an explanation of what the measure is trying to proxy for, particularly as this is an HIV-negative sample. Moreover the variable seems to group those born abroad with those born to immigrants, raising further questions as to what the variables seeks to measure. In addition, the list of countries and regions are not mutually exclusive (e.g. Morocco and Northern Africa are both listed), and the citations listed do not provide any information on region specific HIV prevalence. One might expect a reference citing UNAIDS or other comparable organizations, and potentially a supplementary file clearly listing every country designated as such.

RESULTS

4. The variable name of “sexual orientation” does not seem fully descriptive as it excludes TGW which are part of this sample. The listed subgroups, MSM and MSWM, describe behaviors, not orientations. Likely more consist would be to provide separate variables for gender identity (e.g. male, female, gender non-conforming) and the gender of sexual partners (e.g. male, female, both).

5. A matter of consistency in language, but the authors refer to “latent classes of sexual behaviors” and “latent classes of visits.” This reviewer’s understanding is that the latent classes would refer to those underlying the *population.*

6. Most LCA papers that this author is familiar with provides predicted probability estimates for all included LCA model items, not just a subset. Table 2 would make more sense if it listed probabilities for all the model items.

7. It is this author’s impressions that once the original LCA model confirms the number of latent classes, separate logistic models should be used to estimate associations with key factors.

8. Table 3: what are the aOR results adjusting for? In addition, the p-values seem somewhat redundant to the confidence intervals. If the editor agrees, this reviewer would advocate for retaining only 95% CI.

DISCUSSION

9. Selection bias strikes this author as a somewhat key limitation of the analysis. By using clinical records, this sample presumably over represents the subset of non-PrEP users with more frequent health seeking behaviors. The bias generated by this decision is likely exacerbated by the authors’ decision to include multiple visits per person. Regardless, a discussion of how the authors believe their sampling decisions affects their results would be helpful.

6. PLOS authors have the option to publish the peer review history of their article (what does this mean?). If published, this will include your full peer review and any attached files.

Reviewer #1: No

Reviewer #2: No

---

## [Author Response · Author response to Decision Letter 0]

13 Feb 2023

EDITOR REMARKS

1. I particularly share the concern by one of the reviewers that your analysis could have been biased the fact that multiple observations (visits) were included for the same participant. I suggest to conduct some kind of sensitivity analysis, e.g., by only using one random visit from each study participant. 

Thank you for your remark and helpful suggestion. Please refer to our answer to this reviewer’s comment below:

This is an important point. To be clear, we did correct the standard errors to be independently and identically distributed across participants (and not visits). In this sense, the latent structure should consequently represent that of the population and not study visits. The number of observations per individual were fairly homogeneous in our study (median=2; IQR=1-4, range 1-15). When differences in repeated measurements are observed between individuals, any bias from over sampled individuals (i.e., those with more measurements) would be strongest when their behaviors are consistent across visits. Our experience and previous research shows that at sexual health centers, behaviors tend to vary within individuals over time [1]. Taken together, we would not suspect that differences in number of measurements would bias our results. Nevertheless, in order to address this concern, we have conducted an additional sensitivity analysis where the original LCA model was repeated using only one random visit from each included individual. The parameter estimates from this analysis were comparable to the main analysis. 

The following sentences were added to the methods (lines 150-152):

“Sensitivity analyses were performed to assess potential bias related to calendar year and to the inclusion of multiple visits per individual. The LCA model was hence repeated (i) for each year, separately, and (ii) using only one randomly selected visit per individual.”

and to the results (lines 282-284):

“In a sensitivity analysis, where the LCA model was repeated using only one randomly selected visit per individual, results were comparable to the original model (S4 Table); indicating that inclusion of multiple visits per participant did not likely bias results.”

2. Also as suggested by the reviewer, please also provide results from a "traditional" risk factor approach (e.g., using logistic regression with any STI as the outcome while adjusting for multiple observations per participant) for complementing the LCA approach.

Please refer to our answer to this reviewer’s comment below:

We appreciate the reviewer’s concern. The major issue with these types of “traditional” risk factor approaches is that individual factors are highly correlated, and analytical choices on which variables to include are subjective, while avoiding confounders that would bring rise to mediation, collider, so-called “M” biases, etc. LCA allows us to analyze characteristics that are more likely to be clustered, rather than as individual factors. We believe that this approach is especially informative for “targeting” groups not-using but perhaps needing PrEP. To clarify this issue, the following sentence has been added to the introduction (lines 74-77):

“Considering that many of the sexual behaviors are highly correlated, making it difficult to construct models examining independent pathways to these outcomes, we used an approach that clustered like behaviors within individuals as latent classes to accomplish these study aims.”

JOURNAL REQUIREMENTS

The manuscript and file names have been edited to meet the style requirements.

More information regarding informed consent, including a clarification that we did not include minors, has been provided and the text has been edited accordingly (lines 89-91):

“All individuals visiting an SHC were of age and provided both verbal informed consent and an opt-out option for sharing data with the RIVM, documented by the SHC. All collected data are coded, secured and fully pseudonymized in accordance with Dutch privacy legislation.”

3. Thank you for stating the following in the Competing Interests section: "MP obtained unrestricted research grants and speaker/ advisory fees from Gilead Sciences, Abbvie and MSD; all of which were paid to her institute and were unrelated to the current work. EH obtained unrestricted research grants from Gilead Sciences, which were paid to her institute and were unrelated to the current work. The other authors report no conflicts of interest.". Please confirm that this does not alter your adherence to all PLOS ONE policies on sharing data and materials, by including the following statement: "This does not alter our adherence to PLOS ONE policies on sharing data and materials.”. If there are restrictions on sharing of data and/or materials, please state these. Please note that we cannot proceed with consideration of your article until this information has been declared. Please include your updated Competing Interests statement in your cover letter; we will change the online submission form on your behalf.

The statement “This does not alter our adherence to PLOS ONE policies on sharing data and materials” has been added to the conflict-of-interest statement (lines 403-404) and, as requested, the updated version has been included in the cover letter above.

The data used in this study has been provided by the RIVM, who only allow data sharing upon request and after approval by the registration committee. The data availability statement has been edited to clarify this (lines 4011-414):

“The data used in this study have been provided from the RIVM, where, due to ethical and legal restrictions, including the sensitivity of the information, data sharing is only allowed upon specific request and after approval by the registration committee. Please contact soap@rivm.nl for more information and/or submission requests.” 

4. We note that you have indicated that data from this study are available upon request. PLOS only allows data to be available upon request if there are legal or ethical restrictions on sharing data publicly. In your revised cover letter, please address the following prompts: a) If there are ethical or legal restrictions on sharing a de-identified data set, please explain them in detail (e.g., data contain potentially sensitive information, data are owned by a third-party organization, etc.) and who has imposed them (e.g., an ethics committee). Please also provide contact information for a data access committee, ethics committee, or other institutional body to which data requests may be sent. b) If there are no restrictions, please upload the minimal anonymized data set necessary to replicate your study findings as either Supporting Information files or to a stable, public repository and provide us with the relevant URLs, DOIs, or accession numbers. We will update your Data Availability statement on your behalf to reflect the information you provide.

The data availability statement has been edited to clarify the previously missing or unclear information in lines 411-414):

“The data used in this study have been provided from the RIVM, where, due to ethical and legal restrictions, including the sensitivity of the information, data sharing is only allowed upon specific request and after approval by the registration committee. Please contact soap@rivm.nl for more information and/or submission requests.” 

These analyses were omitted in the revised manuscript; hence this requirement is no longer needed. 

6. Please include captions for your Supporting Information files at the end of your manuscript, and update any in-text citations to match accordingly. 

Captions for supporting information have been added to the last page of the manuscript, according to the provided manuscript body formatting guidelines.

REVIEWER 1

1. last paragraph before method section: You aim to identify subgroups of MSM, MSMW and TGP eligible for but not using PrEP. What I was missing here was migrants from endemic areas. I suppose that also in the NL there are many migrants from Sub-Saharan Africa that do not fall in any of the categories above and still carry a high risk for HIV (i.e., by having heterosexual sex with migrants from same region). Why were they not taken into account? Do they not attend the SHC? Are iv drug users not considered to be a group needing PrEP? Are they in a different health setting?

Good point. To be clear, the focus of this paper was on MSM, MSMW and TGP, which are the groups where the most HIV infections are diagnosed in the Netherlands[2]. Even though worldwide, heterosexual migrants and people who inject drugs are also key groups, they are not the main target population for PrEP and HIV prevention in the Netherlands as HIV incidence among them is relatively low[2, 3]; making the inclusion of these groups fall beyond the scope of this paper.

In addition, if we were to broaden our scope to include heterosexual migrants, data on migrant status are unfortunately limited (and migrants do attend the SHC). In our analyses, we used the official RIVM classification for originating from an STI/HIV endemic area, which does include migrants from Sub-Saharan Africa. However, we are unable to provide any further information on the region of origin due to the pseudonymization of data and as a result of privacy legislation in the Netherlands (GDPR), which ensures individuals are not traceable. We have added the following point to the discussion to reflect on this matter (lines 333-341):

“Individuals from STI/HIV endemic regions were also more present in group 1 compared to other groups. This is despite their potential risk for HIV, as shown in research indicating that the majority of HIV acquisition among MSM migrants from Sub-Saharan Africa and Latin America/Caribbean are postmigration HIV infections, acquired in the European host country [29]. Nonetheless, those from STI/HIV endemic regions that were more often classified in group 1, represents individuals from heterogenous geographical locations. Given that we did not have access to specific information on the origin of these individuals or the lower risk of STI/HIV in the group with a larger proportion of migrants from endemic regions, this becomes difficult to interpret but important to consider for future research.”

We agree that people who inject drugs should be eligible for PrEP based on their high risk of acquiring HIV. Yet, if we were to specifically include these individuals into the scope of our paper, data on injection drug use were largely missing and hence could not be reliably included in analysis. The following sentence has been added to the discussion to clarify this issue (lines 318-320):

“Moreover, we have limited data on whether chemsex includes injecting drug use due to substantial missing data. However, because it poses an increased risk for HIV acquisition, it is worth mentioning as another behavioral characteristic to consider for PrEP uptake [27].”

2. page 4, 1st paragraph: was the sexual behavior self-reported or reported by physicians?

The sexual behavior in this study is all self-reported. The following sentence has been edited to clarify this (lines 84-86):

“Collected data pertain to STI and HIV testing and diagnosis, self-reported sexual behavior, and additional sociodemographic and sexual health-related characteristics.”

3. Page 5, last paragraph: why did you exclude the period of summer 2021?

This period has been excluded because data were not yet available upon data request. This has been clarified in the Methods (lines 86-87):

“Use of data was requested for analysis in July 2021, which pertains to data collected from July 2019 until June 2021.”

4. Again, people from STI/HIV endemic regions are more present in group 1 and thus deprioritized. Although this is what the model is suggesting, I would have liked to see this critically discussed in more depth. People from STI/HIV endemic areas are a very heterogenous group and I doubt that they all carry the same high risk for HIV. You do not necessarily have many STI in this group and still, they may have a high risk for HIV with partners from endemic regions. Among these, are there any migrant subgroups that are bigger than others, have a higher risk for HIV? If so, I wonder if you have tried to look at them separately?

Unfortunately, our data on migrants is limited and although we have included the official RIVM classification for originating from an STI/HIV endemic area, all migrant groups that fall under this classification have been grouped into one. Because of this, we could not provide insight into specific sub-groups in terms of characteristics and HIV risk. 

We have added the following point to the discussion to reflect on this limitation (lines 333-341):

“Individuals from STI/HIV endemic regions were also more present in group 1 compared to other groups. This is despite their potential risk for HIV, as shown in research indicating that the majority of HIV acquisition among MSM migrants from Sub-Saharan Africa and Latin America/Caribbean are postmigration HIV infections, acquired in the European host country [29]. Nonetheless, those from STI/HIV endemic regions that were more often classified in group 1, represents individuals from heterogenous geographical locations. Given that we did not have access to specific information on the origin of these individuals or the lower risk of STI/HIV in the group with a larger proportion of migrants from endemic regions, this becomes difficult to interpret but important to consider for future research.”

5. Page 12, last paragraph: "We found that when many sexual partners, group sex, sex work and chemsex are reported, extra attention to encourage PrEP uptake is warranted" - Should it not be "sex work OR chemsex"? I understood that one of the cited factors would be sufficient to warrant stronger encouragement to uptake PrEP. I suppose it is quite rare to have all factors present in one person.

It is true that any one of these factors would be sufficient to warrant PrEP use. However, with the use of latent classes, it is important to note that the combination of these is associated with the increased risk of STI. The text has been edited accordingly (lines 382-383):

“We found that when many sexual partners, group sex, sex work or chemsex are reported, particularly in combination, extra attention to encourage PrEP uptake is warranted.”

6. It would have been nice to make more concrete recommendations on how to prioritize the groups. How should counsellors proceed to make a concise choice on who to approach? Are you recommending a scoring system?

Thank you for this suggestion, we have added the following recommendation to our conclusion (lines 383-386):

“Prioritizing individuals with one or more of these characteristics may be attained through targeted information provision and counseling. In the case of restricted access and waitlists, an assessment for PrEP need based on the combination of these characteristics may be beneficial.”

7. Please mention what would be foreseen what iv drug users.

To be clear, we refer to injecting drug use during chemsex among mostly MSM. This has now been clarified in the discussion (lines 318-320): 

“Moreover, we have limited data on whether chemsex includes injecting drug use due to substantial missing data. However, because it poses an increased risk for HIV acquisition, it is worth mentioning as another behavioral characteristic to consider for PrEP uptake [27].”

REVIEWER 2

1. Given the abundant existing literature on PrEP uptake (and non-uptake), the introduction could benefit from a clearer justification for using an LCA approach, as opposed to say, a traditional risk factor analysis. What new or additional insights do the authors expect to gain using this technique?

Because individual factors are highly correlated, we would never get “independent” risk-factors with traditional risk factor analysis. Also, LCA allows us to analyze group characteristics rather than individual factors, which is especially informative for our study as we particularly aimed to identify “target” groups not-using but perhaps needing PrEP. To clarify this, the following sentence has been added to the introduction (lines 74-77):

“Considering that many of the sexual behaviors are highly correlated, making it difficult to construct models examining independent pathways to these outcomes, we used an approach that clustered like behaviors within individuals as latent classes to accomplish these study aims.”

2. In this reviewer’s understanding, LCA models decipher latent structure underlying a population, not their visit patterns. Therefore, the authors’ decision to include data from multiple visits from the same subjects seems as though it would falsely inflate the dataset. This likely over represents the experiences of those with more study visits, weighting results that skew towards their characteristics.

Even though behavioral patterns change over time, thereby lessening the influence of repeated observations, you make an important point. In order to address your concern that our analyses may have been biased due to the use of multiple visits per individual, we have conducted an additional sensitivity analysis where the original LCA model is repeated using only one random visit from each included individual. There is little difference between both models. Hence, although the bias you mention is a valid point, it is not a particular issue in our study.

The following sentences were added to the methods (lines 150-152):

“Sensitivity analyses were performed to assess potential bias related to calendar year and to the inclusion of multiple visits per individual. The LCA model was hence repeated (i) for each year, separately, and (ii) using only one randomly selected visit per individual.”

and to the results (lines 282-284):

“In a sensitivity analysis, where the LCA model was repeated using only one randomly selected visit per individual, results were comparable to the original model (S4 Table); indicating that inclusion of multiple visits per participant did not likely bias results.”

3. A clearer justification for the authors’ decision to code for whether a participant is from an HIV endemic setting would be helpful. This could include an explanation of what the measure is trying to proxy for, particularly as this is an HIV-negative sample. Moreover, the variable seems to group those born abroad with those born to immigrants, raising further questions as to what the variables seeks to measure. In addition, the list of countries and regions are not mutually exclusive (e.g., Morocco and Northern Africa are both listed), and the citations listed do not provide any information on region specific HIV prevalence. One might expect a reference citing UNAIDS or other comparable organizations, and potentially a supplementary file clearly listing every country designated as such.

We thank the reviewer for raising these points, which were also discussed by Reviewer 1 (please refer to comments 1 and 4). Unfortunately, our data on migrants are limited. In our analyses, we have included the official RIVM classification for originating from an STI/HIV endemic area because only these data were available to us. Due to the pseudonymization of data and as a result of privacy legislation in the Netherlands (GDPR), specific countries cannot be further specified to avoid traceability of participants. The references refer to the RIVM definition and the source they use (including the UNAIDS guidelines), hence we do not want to attribute it to external references. 

We realize that this definition represents individuals from heterogenous geographical regions, yet it does provide some insight into the role that migration may play when involving countries that have an increased HIV prevalence. As a result, we did not make any strong conclusions pertaining to this determinant. Also, in line with reviewer 1’s comments, we have included the following statement to the discussion (lines 333-341):

“Individuals from STI/HIV endemic regions were also more present in group 1 compared to other groups. This is despite their potential risk for HIV, as shown in research indicating that the majority of HIV acquisition among MSM migrants from Sub-Saharan Africa and Latin America/Caribbean are postmigration HIV infections, acquired in the European host country [29]. Nonetheless, those from STI/HIV endemic regions that were more often classified in group 1, represents individuals from heterogenous geographical locations. Given that we did not have access to specific information on the origin of these individuals or the lower risk of STI/HIV in the group with a larger proportion of migrants from endemic regions, this becomes difficult to interpret but important to consider for future research.”

Morocco is indeed redundant when northern Africa is already listed, it has hence been removed in the text and table footnotes. 

4. The variable name of “sexual orientation” does not seem fully descriptive as it excludes TGW which are part of this sample. The listed subgroups, MSM and MSWM, describe behaviors, not orientations. Likely more consist would be to provide separate variables for gender identity (e.g., male, female, gender non-conforming) and the gender of sexual partners (e.g., male, female, both).

Agreed and thank you for your suggestions. We have changed the variable name to “sexual partner(s)” and the category names to “male” and “male and female”. This change has been made in all tables, footnotes and text. This is in accordance with the way data was collected, as individuals were only asked whether their sex partners were male, female or both (and we excluded those only answering “female”, leaving us with the categories “male” and “male and female”). 

5. A matter of consistency in language, but the authors refer to “latent classes of sexual behaviors” and “latent classes of visits.” This reviewer’s understanding is that the latent classes would refer to those underlying the *population*.

Our apologies for any confusion. Latent classes represent an ensemble of random variables, which are derived from observations during visits. The reviewer is right in that the usage here is inconsistent (and “latent classes of visits” is not entirely appropriate). We have now ensured that “latent classes of sexual behavior” is used throughout the manuscript.

6. Most LCA papers that this author is familiar with provides predicted probability estimates for all included LCA model items, not just a subset. Table 2 would make more sense if it listed probabilities for all the model items.

Our apologies. We are not entirely clear what the reviewer is referring to with the term “subset”. The model has a latent variable component and later, a set of covariates that are regressed on the latent variable component. Predicted probability estimates are certainly used for all items used in the latent class component (which is the case for Table 2). 

Perhaps this was unclear, hence we have clarified which variables were used in the latent portion of the model (along with their distribution) (lines 123-128):

“We used a generalized structural equation modelling approach [18]. Briefly, a latent variable model was constructed, based solely on sexual behavior variables, where the probability of engaging in each behavior given latent class k was modelled by an intercept, αdk, specific to each sexual behavior variable d and class k. Sexual behavior characteristics were modeled as dichotomous variables: number of partners (≥6/≤5; cut-off based on median), chemsex (yes/no), group sex (yes/no) and sex work (yes/no).” 

7. It is this author’s impressions that once the original LCA model confirms the number of latent classes, separate logistic models should be used to estimate associations with key factors.

We assume that the reviewer is referring to an analysis in which, after the number of latent classes are confirmed, each individual is assigned to a latent class based on the highest a posteriori probability of latent class membership and a separate logistic regression model is run using these assigned classes. This approach has certainly been used by others and indeed, we did use a posteriori probabilities to produce numerical distributions of study population characteristics (i.e., Table 1). 

The issue with this approach is that parameter estimates are biased by misclassifying latent classes (i.e., some individuals are assigned to an incorrect latent class). To avoid this issue, the association between the latent class and any given covariate can be directly modeled through some maximum likelihood approach (in this case, the mathematical equivalent of a structural equation model).

We preferred not to give an overly technical explanation of this modeling choice. Instead, we have cited a reference where this type of modeling is more extensively addressed and have made the following edits to the methods (lines 123-128): 

“We used a generalized structural equation modelling approach [18]. Briefly, a latent variable model was constructed, based solely on sexual behavior variables, where the probability of engaging in each behavior given latent class k was modelled by an intercept, αdk, specific to each sexual behavior variable d and class k. Sexual behavior characteristics were modeled as dichotomous variables: number of partners (≥6/≤5; cut-off based on median), chemsex (yes/no), group sex (yes/no) and sex work (yes/no).” 

We also tried to make our modeling approach clearer in the footnote of Table 3 (lines 271-272):

“Parameter estimates for the associations between STI diagnosis or sociodemographic variables and latent classes were directly obtained from a generalized structural equation model.”

8. Table 3: what are the aOR results adjusting for? In addition, the p-values seem somewhat redundant to the confidence intervals. If the editor agrees, this reviewer would advocate for retaining only 95% CI.

The OR is adjusted for all variables present in the table. To clarify this, the following footnote has been added to the respective tables:

“*aOR: all models were adjusted for the variables present in the table.”

We agree that p-values are redundant and have been removed from the tables and text. 

9. Selection bias strikes this author as a somewhat key limitation of the analysis. By using clinical records, this sample presumably over represents the subset of non-PrEP users with more frequent health seeking behaviors. The bias generated by this decision is likely exacerbated by the authors’ decision to include multiple visits per person. Regardless, a discussion of how the authors believe their sampling decisions affects their results would be helpful.

First, we acknowledge that by using data from the SHCs, we are studying a population with more frequent health care seeking behavior. Those who seek sexual healthcare are likely engaging in behaviors associated with higher HIV risk (i.e., reflecting likely PrEP eligibility), and by being in contact with care, they reflect a population that “could” have been on PrEP through the SHCs had they wanted to and had availability been unrestricted. The question our study aimed to answer (i.e., who is eligible for but not using PrEP) required information on the characteristic and sexual behavior of these individuals; and the data from the SHC’s provides this information. 

Second, our decision to include multiple visits per individual has had minimal impact on our analyses, as the sensitivity analysis that was added in response to your comment Nr. 2, showed that the LCA model using one randomly selected visit per individual had comparable results to the original model. The bias you mention is certainly a valid concern, but not a particular issue in our study. Accordingly, the following sentences were added to the methods (lines 150-152):

“Sensitivity analyses were performed to assess potential bias related to calendar year and to the inclusion of multiple visits per individual. The LCA model was hence repeated (i) for each year, separately, and (ii) using only one randomly selected visit per individual.”

and to the results (lines 282-284):

“In a sensitivity analysis, where the LCA model was repeated using only one randomly selected visit per individual, results were comparable to the original model (S4 Table); indicating that inclusion of multiple visits per participant did not likely bias results.”

References used in author response: 

1. Basten MGJ, van Wees DA, Matser A, Boyd A, Rozhnova G, den Daas C, et al. Time for change: Transitions between HIV risk levels and determinants of behavior change in men who have sex with men. PLOS ONE. 2021;16(12):e0259913.

2. van Sighem AI, Wit FWNM, Boyd A, Smit C, Matser A, van der Valk M. HIV Monitoring Report 2022. Amsterdam: stichting hiv monitoring; 2022.

3. van Wees D, Visser M, van Aar F, de Coul EO, Staritsky L, Sarink D, et al. Sexually transmitted infections in the Netherlands in 2021. RIVM rapport 2022-0023. 2022.

---

## [Decision Letter · Decision Letter 1]

22 Mar 2023

Characterizing subgroups of sexual behaviors among men who have sex with men eligible for, but not using, PrEP in the Netherlands

PONE-D-22-30781R1

Dear Dr. de la Court,

We’re pleased to inform you that your manuscript has been judged scientifically suitable for publication and will be formally accepted for publication once it meets all outstanding technical requirements.

Kind regards,

Joel Mossong, PhD

Academic Editor

PLOS ONE

Additional Editor Comments (optional):

Reviewers' comments:

Reviewer's Responses to Questions

**Comments to the Author**

1. If the authors have adequately addressed your comments raised in a previous round of review and you feel that this manuscript is now acceptable for publication, you may indicate that here to bypass the “Comments to the Author” section, enter your conflict of interest statement in the “Confidential to Editor” section, and submit your "Accept" recommendation.

Reviewer #1: All comments have been addressed

Reviewer #2: (No Response)

2. Is the manuscript technically sound, and do the data support the conclusions?

Reviewer #1: Yes

Reviewer #2: Partly

3. Has the statistical analysis been performed appropriately and rigorously? 

Reviewer #1: Yes

Reviewer #2: N/A

4. Have the authors made all data underlying the findings in their manuscript fully available?

Reviewer #1: No

Reviewer #2: No

5. Is the manuscript presented in an intelligible fashion and written in standard English?

Reviewer #1: Yes

Reviewer #2: Yes

6. Review Comments to the Author

Reviewer #1: All my comments have been addressed, thank you. The limitations of the analysis hs been addressed in the manuscript.

Reviewer #2: I appreciate the author's work to incorporate a sensitivity analysis to address earlier mentioned concerns about an LCA that uses multiple data points per person. But retaining the original approach in the paper seems to ignore the fundamental limitations of this approach. This reviewer's recommendation is to fully replace the original model (and associated results) with that of the version performed for the sensitivity analysis.

Additionally, I share the authors' curiosity about potential latent classes emergent in clients' time-dependent sexual behaviors. But if this was indeed the primary interest, perhaps a latent trajectory analysis would be the more appropriate analytical approach. This is not a requirement but just a comment for the authors to consider for future analyses.

7. PLOS authors have the option to publish the peer review history of their article (what does this mean?). If published, this will include your full peer review and any attached files.

Reviewer #1: No

Reviewer #2: No

---

## [Editor Report · Acceptance letter]

28 Mar 2023

PONE-D-22-30781R1 

Characterizing subgroups of sexual behaviors among men who have sex with men eligible for, but not using, PrEP in the Netherlands 

Dear Dr. de la Court:

I'm pleased to inform you that your manuscript has been deemed suitable for publication in PLOS ONE. Congratulations! Your manuscript is now with our production department. 

Kind regards, 

on behalf of

Dr. Joel Mossong 

Academic Editor

PLOS ONE